# Prognosticators for Patients with Pancreatic Ductal Adenocarcinoma Who Received Neoadjuvant FOLFIRINOX or Gemcitabine/Nab-Paclitaxel Therapy and Pancreatectomy

**DOI:** 10.3390/cancers15092608

**Published:** 2023-05-04

**Authors:** Yi Tat Tong, Zongshan Lai, Matthew H. G. Katz, Laura R Prakash, Hua Wang, Deyali Chatterjee, Michael Kim, Ching-Wei D. Tzeng, Jeffrey E. Lee, Naruhiko Ikoma, Asif Rashid, Robert A. Wolff, Dan Zhao, Eugene J. Koay, Anirban Maitra, Huamin Wang

**Affiliations:** 1Department of Pathology, University of Texas MD Anderson Cancer Center, 1515 Holcombe Blvd, Houston, TX 77030, USA; 2Department of Surgical Oncology, University of Texas MD Anderson Cancer Center, 1515 Holcombe Blvd, Houston, TX 77030, USA; 3Department Gastrointestinal Medical Oncology, University of Texas MD Anderson Cancer Center, 1515 Holcombe Blvd, Houston, TX 77030, USA; 4Department of Radiation Oncology, University of Texas MD Anderson Cancer Center, 1515 Holcombe Blvd, Houston, TX 77030, USA; 5Department of Translational Molecular Pathology, University of Texas MD Anderson Cancer Center, 1515 Holcombe Blvd, Houston, TX 77030, USA

**Keywords:** pancreatic cancer, neoadjuvant therapy, FOLFIRINOX, gemcitabine/nab-paclitaxel, tumor response grade, tumor stage, lymph node metastasis, survival

## Abstract

**Simple Summary:**

This study is to examine the clinical and pathologic characteristics and survival in patients who received neoadjuvant FOLFINOX or neoadjuvant gemcitabine/nab-paclitaxel (GemNP) followed by surgery with curative intent. Our study demonstrated that neoadjuvant FOLFIRINOX treatment is associated with younger age, higher rate of borderline resectable and locally advance disease, higher rate of radiation, lower ypN stage, and higher frequency of complete or near complete pathologic response compared to the GemNP group, but no significant differences in either disease-free survival or overall survival between these two treatment groups. We also demonstrated that multiple pathologic factors, including tumor response group, ypT, ypN, LVI, PNI, and resection margin status, were significant prognostic factors for survival in this group of PDAC patients. In addition, our findings suggest that the tumor size of 1.0 cm is a better cutoff for ypT2 in PDAC patients who received neoadjuvant therapy.

**Abstract:**

Neoadjuvant FOLFIRINOX and gemcitabine/nab-paclitaxel (GemNP) therapies are increasingly used to treat patients with pancreatic ductal adenocarcinoma (PDAC). However, limited data are available on their clinicopathologic prognosticators. We examined the clinicopathologic factors and survival of 213 PDAC patients who received FOLFIRINOX with 71 patients who received GemNP. The FOLFIRINOX group was younger (*p* < 0.01) and had a higher rate of radiation (*p* = 0.049), higher rate of borderline resectable and locally advanced disease (*p* < 0.001), higher rate of Group 1 response (*p* = 0.045) and lower ypN stage (*p* = 0.03) than the GemNP group. Within FOLFIRINOX group, radiation was associated with decreased lymph node metastasis (*p* = 0.01) and lower ypN stage (*p* = 0.01). The tumor response group, ypT, ypN, LVI and PNI, correlated significantly with both DFS and OS (*p* < 0.05). Patients with the ypT0/T1a/T1b tumor had better DFS (*p* = 0.04) and OS (*p* = 0.03) than those with ypT1c tumor. In multivariate analysis, the tumor response group and ypN were independently prognostic factors for DFS and OS (*p* < 0.05). Our study demonstrated that the FOLFIRINOX group was younger and had a better pathologic response than the GemNP group and that the tumor response group, ypN, ypT, LVI and PNI, are significant prognostic factors for survival in these patients. Our results also suggest that the tumor size of 1.0 cm is a better cut off for ypT2. Our study highlights the importance of systemic pathologic examination and the reporting of post-treatment pancreatectomies.

## 1. Introduction

Pancreatic cancer is the seventh leading cause of global cancer-related deaths [1,2]. In the United States, it ranks the third leading cause of cancer-related deaths after lung cancer and colorectal cancer [3]. Pancreatic ductal adenocarcinoma (PDAC) is one of the most aggressive cancers and is very difficult to diagnose at an early stage. Most patients with PDAC presented with a locally advanced or metastatic disease at the time of diagnosis, which is not suitable for surgical resection [4]. Even for patients who underwent surgical resection with curative intent, tumor recurrence and/or metastasis is common [5]. Therefore, the prognosis of PDAC patients is poor with a 5-year survival rate of 10.8% [6].

The multidisciplinary neoadjuvant therapy approach plays an important role in the treatment of PDAC patients. The potential benefits for the multidisciplinary neoadjuvant therapy approach includes better tolerability, early treatment for micrometastatic disease, reduction of tumor volume, and a better chance for complete tumor resection [4]. Neoadjuvant therapy shows clear clinical benefits and improved survival for PDAC patients with borderline resectable/locally advanced disease [7,8,9,10]. Even for patients with resectable disease, the paradigm is shifting from upfront surgical resection to the multidisciplinary neoadjuvant therapy approach followed by surgery.

For patients with metastatic PDAC, both FOLFIRINOX (5-fluorouracil, leucovorin, oxaliplatin, and irinotecan) or gemcitabine, plus nab-paclitaxel (GemNP), have been shown to improve survival, compared with gemcitabine alone [11,12,13]. Neoadjuvant FOLFIRINOX or GemNP treatment is also increasingly used to treat patients with potentially resectable PDAC, especially for those with borderline resectable disease. Previous studies have shown that patients treated with neoadjuvant FOLFINOX or GemNP prior to the resection of PDAC, had an improved survival and rate of R0 resection [14,15,16,17,18]. A multicenter study by Macedo et al. showed that patients receiving neoadjuvant regimens of either FOLFINOX or GemNP were associated with better pathological and clinical responses and a longer survival (32 months) with the 3- and 5-year OS rates being 46.3% and 30.3%, respectively [18]. However, limited data is available to compare the FOLFINOX and GemNP regimens, the role of radiation in patients receiving these two regimens, and clinicopathologic prognosticators in this group of PDAC patients. In this study, we compared the clinical and pathologic responses and the survival of 213 PDAC patients who received neoadjuvant FOLFINOX with or without radiation, and 71 patients who received neoadjuvant GemNP with or without radiations followed by surgery with curative intent at our institution. Our results demonstrated that the FOLFIRINOX group was younger and had better pathologic response than the GemNP group. We also demonstrated that tumor response group, primary tumor size/stage (ypT), lymph node metastasis (ypN), lymphovascular invasion (LVI), perineural invasion (PNI), and margin status, were significant prognostic factors for survival in these patients. Systemic pathologic examination and reporting of posttreatment pancreatectomies provide significant and meaningful prognostic information for postoperative patient care and survival.

## 2. Materials and Methods

### 2.1. Study Population

This study was approved by the Institutional Review Board of our institution. Our study group consisted of 213 PDAC patients who received FOLFIRINOX with or without radiation, and 71 patients who received the GemNP with or without radiation, followed by pancreatectomy at our institution from January of 2010 to December of 2019. A total of 202 (71.1%) patients received neoadjuvant radiation therapy and 82 (28.9%) received no radiation therapy. To compare the survival in patients with potentially resectable PDAC who received neoadjuvant FOLFIRINOX or GemNP, with those who underwent upfront surgical resection, our study also included 166 PDAC patients who underwent upfront pancreatectomy. The clinicopathologic characteristics of PDAC patients who underwent upfront pancreatectomy are shown in Appendix A. The histologic diagnosis of PDAC was confirmed in all cases.

### 2.2. Pathologic Examination of the Pancreatectomy Specimens

The pathology evaluation of pancreatectomy specimens, including the ypT, ypN, LVI, PNI, and margin status, was performed and reported using standardized protocol established at our institution. To adequately evaluate the post-therapy tumor, the entire tumor/bed was submitted for histologic examination in 185 (65%) patients. The entire pancreas with adjacent tissue was submitted in 51 cases (18%) due to the absence of grossly apparent lesion(s), or because the initial sections revealed no or a minimal amount of viable tumor. The median number of blocks from the tumor and pancreas was 23 (range: 7–73). Histologic tumor response grading was performed using both the MD Anderson grading system and the College of American Pathologists (CAP) grading system [4,19,20]. For statistical analysis, the patients were grouped into Group 1 (complete or near complete response, the MD Anderson or CAP grade 0 or 1) and Group 2 (the MD Anderson grade 2 or CAP grades 2 and 3), since the number of patients with grade 0 (complete pathologic response) was very small (11, 3.9%). The median number of lymph nodes examined was 27 (range: 5–85). Pathologic stages were classified according to the American Joint Committee on Cancer (AJCC) Staging Manual, 8th edition [21]. All cases were reviewed by two gastrointestinal pathologists (Y.T.T and Z.L.) who were uninformed about the clinical and follow up information.

### 2.3. Clinical Data and Follow Up

The clinical data were retrieved from a prospectively maintained database. The clinicopathologic parameters included patients’ gender, age at time of diagnosis, date of diagnosis, clinical classification of the pre-treatment tumor resectability, neoadjuvant therapy regimens and radiation therapy, date and site of tumor recurrence/metastasis (which are diagnosed mainly based on the imaging studies and clinical suspicion during follow up visits), and both date and cause of death if applicable. All clinical and follow up information was verified by reviewing the patient medical number or the U.S. Social Security Index. The median follow-up time was 32.9 months (6.7 to 113.5 months).

### 2.4. Statistical Analysis

The categorical clinicopathologic parameters were correlated using the Chi-square analyses. Independently sampled *t*-tests were used to compare the means between the groups. Survival analyses were performed using the Kaplan–Meier method, in which the log-rank test was used to determine the significance of survival differences among different groups, or univariate and multivariate Cox regression analyses. Disease-free survival (DFS) was calculated from the date of surgery to the date of first recurrence after surgery in patients with recurrence or to the date of last follow-up in patients without recurrence. Overall survival (OS) was calculated from the date of diagnosis to the date of death, or the date of last follow-up if death did not occur. For multivariate survival analysis, the backward stepwise procedure was used to derive the best-fitting Cox proportional hazards models. All clinicopathologic parameters with a *p* value ≤ 0.1 in univariate survival analysis was included in multivariate survival analyses. The statistical analysis was performed using the Statistical Package for Social Sciences software for Windows (Version 26, SPSS, Inc., Chicago, IL, USA). A 2-sided *p* value of less than 0.05 was considered as statistically significant for all statistical analyses.

## 3. Results

### 3.1. Comparison of the Clinical and Pathologic Characteristics and Survival between the FOLFIRINOX Group and the GemNP Group

There were 150 males and 134 females. Patient age ranged from 30 to 85 years with a median of 65.0 years. Before the neoadjuvant therapy, the tumor was clinically classified as potentially resectable, borderline resectable, and locally advanced in 135, 114, and 35 patients, respectively. Pancreatoduodenectomy, distal pancreatectomy and total pancreatectomy were performed in 217, 57 and 10 patients, respectively.

The clinical and pathologic characteristics of two treatment groups are shown in Table 1. The mean age at diagnosis for the FOLFIRINOX group was 62.1 (±8.8) years, which was younger than the 68.1 (±9.7) year for the GemNP group (*p* < 0.001). The FOLFIRINOX group had more borderline resectable or locally advanced disease at the time of diagnosis (62.0% vs. 23.9%, *p* < 0.001). Among 213 patients in the FOLFIRINOX group, 158 (74.2%) patients received neoadjuvant radiation therapy compared to 62.0% (44/71) in GemNP group (*p* = 0.049). The FOLFIRINOX group had more frequent Group 1 response than GemNP group (16.0% vs. 7.0%, *p* = 0.045). The ypN0, ypN1, and ypN2 were present in 50.2%, 34.8% and 15.0%, respectively in the FOLFIRINOX group, compared to 38.0%, 36.6%, and 25.4%, respectively in GemNP group (*p* = 0.03). There was no difference in gender, type of surgery, ypT, margin status, tumor response grade using either the CAP or MD Anderson grading system, and tumor recurrence/metastasis (*p* > 0.05).

The median of the disease-free survival and overall survival of 17.4 months and not reached, respectively, for the FOLFIRINOX group compared to 18.1 months (*p* = 0.53) and 77.2 months (*p* = 0.84), respectively. Within the FOLFIRINOX group, borderline resectable and locally advance disease was present in 113 (71.5%) patients who also received radiation, compared to 29.1% (16/55) for those whose received FOLFIRINOX alone (*p* < 0.001). Among 158 patients who received neoadjuvant FOLFIRINOX with radiation, ypN0, ypN1 and ypN2 were present in 55.1%, 29.1% and 15.8%, respectively, compared to 36.4%, 50.9%, and 12.7%, respectively, among 55 patients who received FOLFIRINOX alone (*p* = 0.01, Table 2). No significant difference was observed for other clinicopathologic parameters, including gender, tumor response grade, ypT, LVI, PNI, margin status and recurrence between the FOLFIRINOX with radiation and FOLFIRINOX alone (*p* > 0.05, Table 2). Similarly, GemNP with radiation group, had a higher rate of borderline resectable and locally advance disease (34.1%, 15/44) than the GemNP group (7.4%, 2/27, *p* = 0.03, Table 2). No significant differences in other clinicopathologic parameters were observed between the GemNP alone group and the GemNP with radiation group (*p* > 0.05).

### 3.2. Tumor Response Grade to Neoadjuvant Therapy Correlates with the Survival

The correlations of tumor response grade using the CAP and the MD Anderson grading systems with survival are shown in Figure 1. Similar to the previous studies, we did not observe significant differences in either disease-free survival or overall survival between patients with the CAP grade 2 response and those with CAP grade 3 response (*p* > 0.05, Figure 1A,B). Using either the CAP or the MD Anderson grading systems, patients can be classified into two prognostic groups: Group 1, the patients the CAP or MD Anderson grade 0 or 1 response (N = 39); and Group 2, the patients with CAP grade 2 or 3 response or MD Anderson grade 2 response (N = 245). The Group 1 patients had better disease-free survival and overall survival than those in Group 2 (*p* < 0.001, Figure 2A,B). The correlations of the tumor response groups with clinicopathologic parameters are shown in Table 3. A total of 34 (16.0%) patients in the FOLFIRINOX group showed group 1 response compared to 7% (5/71) in GemNP group (*p* = 0.045). The tumor response groups correlated significantly with lymphovascular invasion (*p* < 0.001), perineural invasion (*p* < 0.001), ypT stage (*p* < 0.001), ypN stage (*p* < 0.001), margin status (*p* = 0.040), and tumor recurrence (*p* < 0.001).

### 3.3. Correlation of Other Pathologic Parameters with Survival

The ypT correlated significantly with both disease-free survival (*p* = 0.003, Figure 3A) and overall survival (*p* = 0.005, Figure 3B). Although there was no difference in disease-free survival between patients with ypT1a/ypT1b tumor and those with ypT1c tumor (Figure 3A), we found that patients with ypT1a/ypT1b tumor had similar overall survival to those with ypT0 tumor, and furthermore, patients with ypT1c tumor had similar overall survival to those with ypT2 tumor (Figure 3B). Patients with ypT0/T1a/T1b tumor had better disease-free survival (*p* = 0.04) and overall survival (*p* = 0.03) than those with ypT1c (Figure 3C,D).

The median disease-free survival and overall survival for patients with LVI were 12.9 months and 47.6 months, respectively, compared to 19.6 months (*p* = 0.006) and was not reached (*p* = 0.001), respectively, in those without LVI (Figure 4A,B). The median disease-free survival and overall survival for patients with PNI were 14.3 months and 69.3 months, respectively, compared to 29.4 months (*p* = 0.004) and 85.1 months (*p* = 0.01), respectively, in those without PNI (Figure 4C,D). The ypN stage correlated significantly with both disease-free survival (*p* = 0.001) and overall survival (*p* = 0.003, Figure 4E,F). Positive resection margin is associated with shorter disease-free survival (*p* = 0.048), but not overall survival (*p* = 0.15).

Patients with a potentially resectable tumor who received neoadjuvant FOLFIRINOX or GemNP had better disease-free survival (*p* = 0.03) and overall survival (*p* < 0.001) than those who underwent upfront pancreatectomy (Figure 5).

### 3.4. Multivariate Cox Regression Analyses

In multivariate analyses, the tumor response group provided independent prognostic factors for both disease-free survival [hazard ratio (HR): 2.78 (95% CI: 1.59–4.88), *p* < 0.001] and overall survival [HR: 4.13 (95% CI: 1.65–10.35), *p* = 0.002]. The ypN stage was also an independent prognosticator for both disease-free survival (*p* = 0.01) and overall survival (*p* = 0.048). The neoadjuvant therapy group, ypT stage, lymphovascular invasion, perineural invasion and margin status, were not significant for either disease-free survival or overall survival (*p* > 0.05, Table 4).

## 4. Discussion

Neoadjuvant therapies FOLFIRINOX and GemNP are increasingly used to treat patients with potentially resectable PDAC, especially for patients with borderline resectable disease. Limited data is available to directly compare the clinicopathologic factors, the benefits of neoadjuvant radiation therapy, the pathologic outcomes and survival for these two neoadjuvant regimens. In this retrospective study, we compared the clinical and pathologic parameters and survival of 213 PDAC patients who received neoadjuvant FOLFIRINOX with or without radiation to 71 patients who received neoajuvant GemNP with or without radiation therapy. The median disease-free survival was 17.4 months and 18.1 months, respectively, and the median overall survival was 77.2 months and was not reached, respectively, for the FOLFIRINOX and the GemNP group. Both treatment groups had better disease-free survival and overall survival compared to the previously reported results from PDAC patients who received neoadjuvant gemcitabine or 5-fluoruracil-based neoadjuvant therapies [4,8,9,10,22,23,24]. PDAC patents who received neoadjuvant FOLFIRINOX regimen were younger, had more frequent borderline resectable or locally advanced disease at the time of diagnosis, and more frequently received neoadjuvant radiation compared to the GemNP group. However, these differences may be due to the selection bias for patients to receive neoadjuvant FOLFIRINOX versus GemNP since the choice of neoadjuvant chemotherapy is mainly based on the patient’s overall performance status, health condition, underlining comorbidities and patient preferences.

Pathologic examination of the post-treatment pancreatomy specimens showed that neoadjuvant FOLFIRINOX treatment was associated with lower ypN stage and higher frequency of complete or near complete pathologic response (Group 1 response), compared to the GemNP group. However, we did not observe significant differences in either disease-free or overall survival between the FOLFIRINOX group and the GemNP group. Our results are consistent with a recent study, which showed that modified FOLFIRINOX had fewer positive lymph nodes and better treatment response than the GemNP for borderline resectable or locally advanced PDAC patients [25]. Similar to our results, they did not observe a significant difference in overall survival between these two neoadjuvant regimens in their study (*p* = 0.11) [25]. Our results are also consistent with the findings from a recent phase 2 clinical trial, which showed a comparable two-year overall survival for the neoadjuvant mFOLFIRINOX (47%) and GemNP (48%) for PDAC patients with resectable disease [26].

For PDAC patients with resectable and borderline resectable disease, a previous meta-analysis study showed that neoadjuvant FOLFIRINOX plus radiation had better R0 resection rate than neoadjuvant FOLFIRINOX alone. However, the two treatment groups had a similar pooled estimated median survival and pathological outcomes based on the rates of ypN0 and pathologic complete response [27]. Our study showed that neoadjuvant FOLFIRINOX plus radiation was associated with the lower rate of lymph node metastasis, but not with the resection margin status. It is interesting to note the adverse correlation between neoadjuvant radiation and overall survival within the FOLFIRINOX group. Neoadjuvant radiation was not associated with survival in the GemNP group in our study. These results may be confounded with patient selection bias due to the significantly higher rate (71.5%) of borderline resectable and locally advanced disease in the FOLFIRINOX plus radiation group, compared to 29.1% for those who received the FOLFIRINOX alone. The benefits of radiation therapy on survival for patients receiving FOLFIRINOX would be better tested with randomized trials using matched patients. The recent A021501 phase II randomized clinical trial found that patients with borderline resectable PDAC who received neoadjuvant mFOLFIRINOX alone had a better overall survival than those who received mFOLFIRINOX plus hypofractionated radiotherapy [28]. Future randomized trials are needed to determine the benefit of neoadjuvant radiotherapy in PDAC patients with resectable and borderline resectable disease who received neoadjuvant FOLFIRINOX regimen.

The role and benefit of neoadjuvant therapy in patients with resectable PDAC is unclear. In this study, we demonstrated that patients with potentially resectable PDAC who received neoadjuvant FOLFIRINOX or GemNP had better disease-free and overall survival than those who underwent upfront pancreatectomy. Future randomized clinical trials are needed to confirm this important finding.

Similar to our previous studies, we demonstrated that the Group 1 response (complete and near complete pathologic response) in patients who received neoadjuvant FOLFIRINOX or Gem NP, followed by pancreatectomy, was associated with better disease-free survival and overall survival, and correlated with less frequent LVI, PNI, lymph node metastasis, margin positive resection and recurrence, and a lower ypT stage and ypN stage [4,23,29,30]. In contrast, we did not observe significant differences in either disease-free survival or overall survival for patients with CAP grade 2 and those with CAP grade 3 responses, which is consistent with previous studies [19,23,29,31,32,33]. The tumor response group provided an independent prognostic factor for both disease-free and overall survival in multivariate analysis.

The ypN stage and positive lymph node ratio have been shown to be important prognostic factors for PDAC patients who underwent upfront surgical resection, as well as for patients who underwent pancreatectomy after neoadjuvant therapy [24,34,35]. In this study, we demonstrated that the ypN stage was an independent prognostic factor for both disease-free and overall survival in PDAC patients who received neoadjuvant FOLFIRINOX or Gem NP followed by pancreatectomy. In addition, we showed that multiple other pathologic parameters including ypT stage, LVI, PNI and resection margin status are important prognostic factors in this group of PDAC patients. Similar to our results, Chatterjee et al. showed that the ypT stage based on the AJCC 8th edition performed better than the ypT stage based on the AJCC 7th edition in predicting the survival, and that LVI and PNI are important prognosticators for survival in PDAC for patients receiving neoadjuvant therapy and pancreaticoduodenectomy [22,36,37]. It is important to point out that patients with the ypT1a/ypT1b tumor had a similar overall survival to those with ypT0 in this study. On the other hand, patients with the ypT1c tumor had a similar overall survival to those with ypT2 tumor. Patients with the ypT0/T1a/T1b tumor had better disease-free and overall survival than those with ypT1c. Similar results have been reported by Chatterjee et al. in a large cohort of 398 PDAC patients who were treated with neoadjuvant therapy [22]. Based on these results, the tumor size of 1.0 cm is a better cut-off for ypT2 stage in PDAC patients who received neoadjuvant therapy.

The potential limitations of this study include the retrospective nature and potential patient selection bias from a single institution dataset. In addition, our study did not include patients who were treated with neoadjuvant FOLFIRINOX or GemNP with or without radiation, but never making it to surgery. The lack of randomization in this study may preclude drawing a definitive conclusion about the relative efficacy of these two regimens.

## 5. Conclusions

Our study demonstrated that neoadjuvant FOLFIRINOX treatment is associated with younger age, a higher rate of borderline resectable and locally advance disease, higher rate of radiation, lower ypN stage and higher frequency of complete or near complete pathologic response (Group 1 response), compared to the GemNP group; however, there were no significant differences in either disease-free survival or overall survival between these two treatment groups. In addition, our study demonstrated that multiple pathologic factors, including the tumor response group, ypT, ypN, LVI, PNI, and resection margin status, were significant prognostic factors for survival in this group of PDAC patients. Furthermore, our findings suggest that tumor size of 1.0 cm is a better cutoff for ypT2 in PDAC patients who received neoadjuvant therapy. Systemic pathologic examination and reporting of the posttreatment pancreatectomies, provide significant and meaningful prognostic information for postoperative patient care and survival.

## Figures and Tables

**Figure 1 cancers-15-02608-f001:**
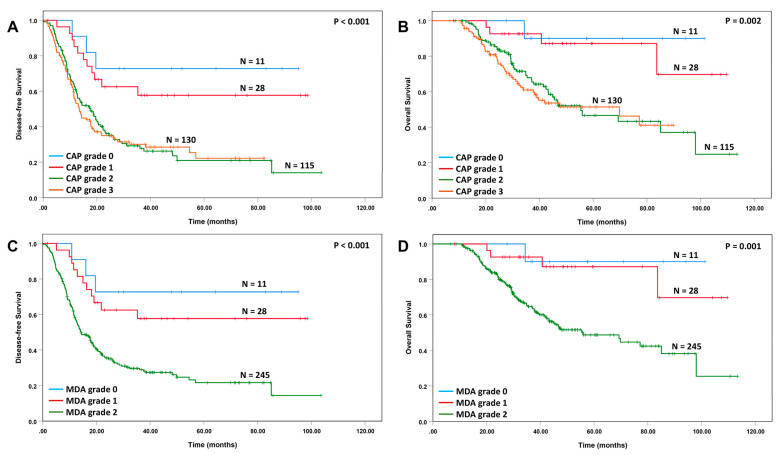
Kaplan–Meier curves for disease-free survival and overall survival stratified by the CAP and MD Anderson tumor response grade. A and B, Kaplan–Meier curves for disease-free survival (**A**) and overall survival (**B**) stratified by the CAP tumor response grade. There are no differences in either disease-free survival or overall survival between CAP grade 2 and CAP grade 3. (**C**,**D**), Kaplan–Meier curves for disease-free survival (**C**) and overall survival (**D**) stratified by the MD Anderson tumor response grade.

**Figure 2 cancers-15-02608-f002:**
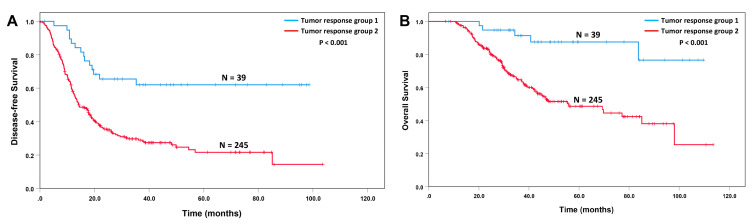
Kaplan–Meier curves for disease-free survival and overall survival stratified by tumor response group. (**A**) Patients with tumor response group 1 have better disease-free survival (*p* < 0.001) and (**B**) overall survival (*p* < 0.001) than those with tumor response group 2.

**Figure 3 cancers-15-02608-f003:**
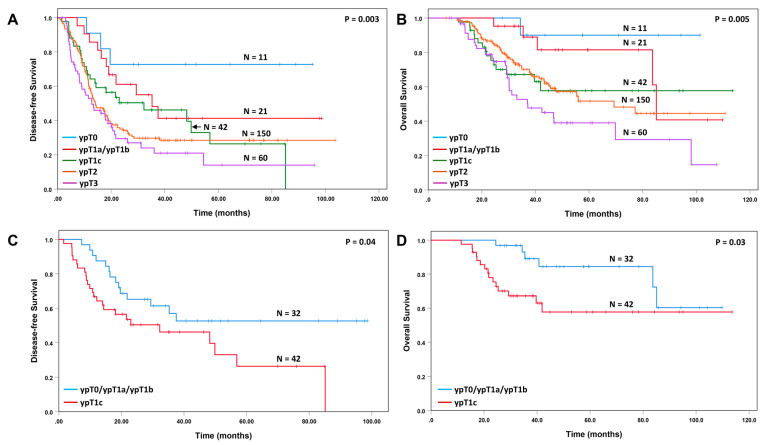
Kaplan–Meier curves for disease-free survival (**A**) and overall survival (**B**) stratified by the ypT stage. Patients with ypT1a/ypT1b tumor have similar overall survival to those with ypT0 tumor, while patients with ypT1c tumor have similar overall survival to those with ypT2 disease. (**C**,**D**), Kaplan–Meier curves for disease-free survival and overall survival for patients with ypT0/ypT1a/ypT1b compared to those with ypT1c tumors. Patients with ypT0/ypT1a/ypT1b have better disease-free survival (*p* = 0.04) and overall survival (*p* = 0.03) than those with ypT1c tumors.

**Figure 4 cancers-15-02608-f004:**
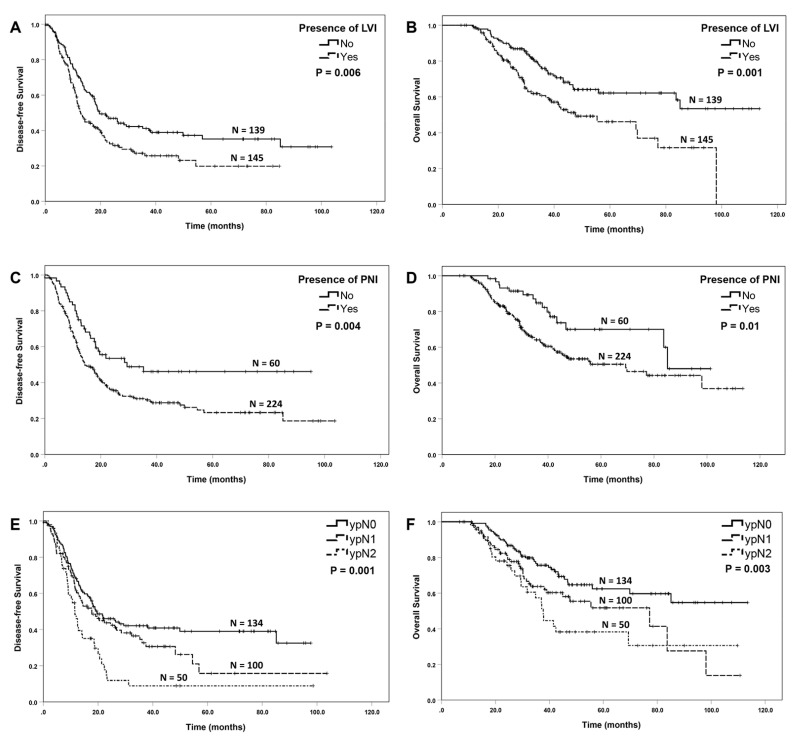
Kaplan–Meier curves for disease-free survival and overall survival stratified by the lymphovascular invasion (LVI, (**A**,**B**)), perineural invasion (PNI, (**C**,**D**)) and ypN stage (**E**,**F**).

**Figure 5 cancers-15-02608-f005:**
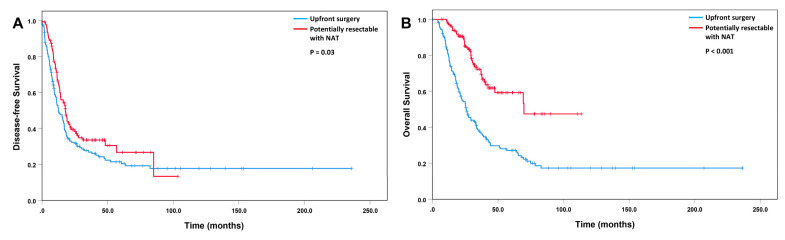
Kaplan–Meier curves for disease-free survival (**A**) and overall survival (**B**) for patients with a potentially resectable tumor who received neoadjuvant FOLFIRINOX or GemNP (NAT, N = 135) compared to patients who underwent upfront surgery (N = 166).

**Table 1 cancers-15-02608-t001:** Comparison of the Clinicopathologic Characteristics Between the FOLFORINOX Group and GemNP Group.

Clinicopathologic Features	FOLFORINOX Group N (%)	GemNP Group N (%)	*p* Values
Gender			0.34
Female	97 (45.5)	37(52.1)	
Male	116 (54.5)	34 (47.9)	
Mean age ± SD (years)	62.1 ± 8.8	68.1 ± 9.7	<0.001
Clinical tumor classification			<0.001
Potentially resectable	81 (38.0)	54 (76.1)	
Borderline resectable	103 (48.4)	11 (15.5)	
Locally advanced	29 (13.6)	6 (8.4)	
Neoadjuvant radiation			0.049
No	55 (25.8)	27 (38.0)	
Yes	158 (74.2)	44 (62.0)	
Type of surgery			0.11
Pancreaticoduodenectomy	158 (74.2)	59 (83.1)	
Distal pancreatectomy	45 (21.1)	12 (16.9)	
Total pancreatectomy	10 (4.7)	0 (0.0)	
Pathologic tumor stage			0.46
ypT0	10 (4.7)	1 (1.4)	
ypT1	48 (22.5)	15 (21.1)	
ypT2	108 (50.7)	42 (59.2)	
ypT3	47 (22.1)	13 (18.3)	
Pathologic lymph node stage			0.03 *
ypN0	107 (50.2)	27 (38.0)	
ypN1	74 (34.8)	26 (36.6)	
ypN2	32 (15.0)	18 (25.4)	
Margin status			0.63
Negative	165 (77.5)	53 (74.6)	
Positive	48 (22.5)	18 (25.4)	
CAP grading			0.29
0	10 (4.7)	1 (1.4)	
1	24 (11.3)	4 (5.6)	
2	96 (45.0)	34 (47.9)	
3	83 (39.0)	32 (45.1)	
MD Anderson grading			0.16
0	10 (4.7)	1 (1.4)	
1	24 (11.3)	4 (5.6)	
2	179 (84.0)	66 (93.0)	
Tumor response group			0.045 **
Group 1	34 (16.0)	5 (7.0)	
Group 2	179 (84.0)	66 (93.0)	
Recurrence			0.67
No	75 (35.2)	27 (38.0)	
Yes	138 (64.8)	44 (62.0)	

Abbreviations: CAP, the College of American Pathologists; GemNP, gemcitabine plus nab-paclitaxel; * *p* value by the Linear-by-linear association; ** *p* value by the Likelihood Ratio.

**Table 2 cancers-15-02608-t002:** Comparison of the Clinicopathologic Characteristics Between Patients with Radiation and Those Without Radiation in FOLFORINOX and Gemcitabine/nab-paclitaxel Groups.

Clinicopathologic Features	FOLFORINOX Alone	FOLFORINOX with RT	*p* Values	GemNP Alone	GemNP with RT	*p* Values
Gender			0.54			0.13
Female	27	70		11	26	
Male	28	88		16	18	
Mean age ± SD (years)	62.0 ± 8.4	62.2 ± 9.0	0.89	66.8 ± 8.9	68.8 ± 10.2	0.40
Clinical tumor classification			<0.001			0.03
Potentially resectable	36	45		25	29	
Borderline resectable	10	93		2	9	
Locally advanced	9	20		0	6	
Type of surgery			0.22			0.35
Pancreaticoduodenectomy	36	122		21	38	
Distal pancreatectomy	16	29		6	6	
Total pancreatectomy	3	7		0	0	
Pathologic tumor stage			0.48			0.42
ypT0	3	7		0	1	
ypT1	14	34		4	11	
ypT2	30	78		16	26	
ypT3	8	39		7	6	
Pathologic lymph node stage			0.01			0.39
ypN0	20	87		8	19	
ypN1	28	46		10	16	
ypN2	7	25		9	9	
Lymphovascular invasion			0.20			0.35
Negative	23	82		11	23	
Positive	32	76		16	21	
Perineural invasion			0.90			0.14
Negative	13	36		2	9	
Positive	42	122		25	35	
Margin status			0.20			0.64
Negative	46	119		21	32	
Positive	9	39		6	12	
Tumor response group			0.45			0.93
Group 1	7	27		2	3	
Group 2	48	131		25	41	
Recurrence			0.23			0.89
No	23	52		10	17	
Yes	32	106		17	27	

Abbreviations: RT, radiation therapy; GemNP, gemcitabine plus nab-paclitaxel.

**Table 3 cancers-15-02608-t003:** Correlations of Tumor Response Group with Clinicopathologic Factors.

Clinicopathologic Factors	Tumor Response Group		*p* Values
	Group 1 N (%)	Group 2 N (%)	
Gender			0.37
Female	21 (15.7)	113 (84.3)	
Male	18 (12.0)	132 (88.0)	
Mean age ± SD (years)	58.1 ± 10.2	64.5 ± 9.0	<0.001
Neoadjuvant therapy group			0.045 *
FOLFIRINOX	34 (16.0)	179 (84.0)	
GemNP	5 (7.0)	66 (93.0)	
Neoadjuvant radiation			0.39
No	9 (11.0)	73 (89.0)	
Yes	30 (14.9)	172 (85.1)	
Lymphovascular invasion			<0.001
Negative	36 (25.9)	103 (74.1)	
Positive	3 (2.1)	142 (97.9)	
Perineural invasion			<0.001
Negative	28 (46.7)	32 (53.3)	
Positive	11 (4.9)	213 (95.1)	
Margin status			0.04
Negative	35 (16.1)	183 (83.9)	
Positive	4 (6.1)	62 (93.9)	
ypT stage			<0.001
ypT0	11 (100)	0 (0.0)	
ypT1a or 1b	15 (71.4)	6 (28.6)	
ypT1c	5 (11.9)	37 (88.1)	
ypT2	6 (4.0)	144 (96.0)	
ypT3	2 (3.3)	58 (96.7)	
ypN stage			<0.001
ypN0	32 (23.9)	102 (76.1)	
ypN1	5 (5.0)	95 (95.0)	
ypN2	2 (4.0)	48 (96.0)	
Recurrence			<0.001
No	25 (24.5)	77 (75.5)	
Yes	14 (7.7)	168 (92.3)	

* *p* value by the Likelihood Ratio.

**Table 4 cancers-15-02608-t004:** Multivariate Cox Regression Analysis of Disease-free Survival and Overall Survival.

Characteristic	No. of Patients	Disease-Free Survival		Overall Survival	
		HR (95% CI)	*p* Value	HR (95% CI)	*p* Value
Neoadjuvant therapy group					
FOLFORINOX (reference)	213	1.00		1.0	
GemNP	71	0.80 (0.57–1.13)	0.21	0.89 (0.56–1.42)	0.62
Perineural invasion					
No (reference)	60	1.00		1.0	
Yes	224	0.96 (0.60–1.52)	0.86	0.93 (0.50–1.74)	0.82
Lymphovascular invasion					
No (reference)	139	1.00		1.00	
Yes	145	1.09 (0.79–1.52)	0.59	1.34 (0.87–2.07)	0.18
Margin status					
Negative (reference)	218				
Positive	66	1.07 (0.75–1.54)	0.71	NA	NA
ypT stage			0.56		0.31
ypT0 (reference)	11	1.00		1.00	
ypT1	63	1.59 (0.44–5.75)	0.48	1.69 (0.19–15.23)	0.64
ypT2	150	1.77 (0.48–6.62)	0.39	1.46 (0.16–13.69)	0.74
ypT3	60	2.11 (0.55–8.08)	0.27	2.24 (0.24–21.35)	0.48
ypN stage			0.01		0.048
ypN0 (reference)	134	1.00		1.00	
ypN1	100	1.08 (0.77–1.51)	0.66	1.35 (0.86–2.12)	0.19
ypN2	50	1.77 (1.20–2.62)	0.004	1.90 (1.14–3.17)	0.01
Tumor response group					
Group 1 (reference)	39	1.00		1.00	
Group 2	245	2.78 (1.59–4.88)	<0.001	4.13 (1.65–10.35)	0.002

Abbreviations: HR, Hazard ratio; CI, confidence interval; NA, not applicable.

## Data Availability

The data is unavailable due to privacy or ethical restrictions.

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
