# Peer review of "Prognosticators for Patients with Pancreatic Ductal Adenocarcinoma Who Received Neoadjuvant FOLFIRINOX or Gemcitabine/Nab-Paclitaxel Therapy and Pancreatectomy"

_cancers, 2023, doi:10.3390/cancers15092608_

Round 1

Reviewer 1 Report

The authors studied the clinical and pathologic characteristics and survival in PDAC patients who received neoadjuvant FOLFINOX and patients who received neoadjuvant gemcitabine/nab-paclitaxel (GemNP) followed by surgery with curative intent. The authors demonstrated that neoadjuvant FOLFIRINOX treatment is associated with younger age, higher rate of borderline resectable and locally advance disease, higher rate of radiation, lower ypN stage and higher frequency of complete or near complete pathologic response compared to the GemNP group, but no significant differences in either disease-free survival or overall survival between these two treatment groups. The authors also demonstrated that multiple pathologic factors, including tumor response group, ypT, ypN, LVI, PNI, and resection margin status were significant prognostic factors for survival in this group of PDAC patients. In addition, the authors proposed tumor size of 1.0 cm is a better cutoff for ypT2 in PDAC patients who received neoadjuvant therapy.

Comments:

1.      The authors gave the clinical tumor classification as potentially resectable, borderline resectable and locally advanced 3 groups. How about the pre-treatment tumor staging information, such as cTNM? This is very important information to compare the two treatment groups. Please add the pretreatment cTNM information.

2.      The data showed that the neoadjuvant FOLFIRINOX treatment is associated with younger age, higher rate of borderline resectable and locally advance disease, higher rate of radiation, lower ypN stage and higher frequency of complete or near complete pathologic response compared to the GemNP group. How these patients were decided to receive different treatment and divided into different groups. If there was a chosen bias initially to put patients into different groups, the comparison will be invalid. Please explain how the patients were divided into these two groups.

3.      When the authors have the cTNM data, they can compare these two treatment groups stage by stage. In this way, they will conclude if there is any treatment effect between these 2 groups.

Author Response

We thank the reviewer for reviewing our manuscript and for the valuable comments on our manuscript. Please see the attached revision letter for our responses to the comments.

Reviewer 2 Report

Tong et al. described a heterogenous cohort of pancreatic cancer patients who had neoadjuvant chemotherapy either with FOLFIRINOX or Gemcitabine/nab-paclitaxel and with or without additional radiotherapy. The authors tried to then draw comparisons between performance of the respective chemotherapy regimens and identify important clinicopathological feature associations. They then reported that ypT1c had similar survival to T2 than ypT1a/ypT1b.

The heterogeneity in the population makes interpretation of their results difficult because upfront resectable clearly have a different prognostic outlook compared to the borderline and locally advanced. Whilst I appreciate that they are being inclusive to increase number of cases to do some of the multivariate analyses but at least do a subgroup analysis of the upfront resectable only or case matched control of patients treated with FOLFIRINOX +/- radiation to answer specific unanswered questions in the field – whether upfront neoadjuvant therapy (not even certain if it was total neoadjuvant or more perioperative in duration) is better than simply upfront resection. Secondly, what is the role of additional radiotherapy in those that were not resectable post neoadjuvant chemotherapy. The numbers will be small but at least would make the study more novel. The case matching can easily be done within the same statistical software you are using.

Tong et al. have chosen to define the outcome of the neoadjuvant therapy based from the date of surgery, this does not take into account the difference in the neoadjuvant period and excluded those that do not make it to surgery. This exclusion and arbitrary endpoint definition at least partially explain the reported “better” survival outcomes compared to the literature and makes interpretation of their results difficult and potentially misleading for readers who do not pay close attention to the detail. I would favour redefining the study endpoints definition for consistency of reporting and external validity. I think better references for the survival of the neoadjuvant cohort should come from published meta-analyses rather than single institution-reported series from authors’ past publication https://pubmed.ncbi.nlm.nih.gov/29708592/. https://www.ncbi.nlm.nih.gov/pmc/articles/PMC7231310/.

The interesting observation I’ve made is that there was neoadjuvant radiation been used in the potentially resectable group. It remains unclear to me whether this group that had the concurrent radiation represent the uncommon practice of radiation to upfront resectable patients (and for what rationale) or as the group name suggests, these patients’ cancer had other undesirable features that made them only potentially resectable and thus differs to the convention of upfront resectable. My apologies if I have misinterpreted this but otherwise the heterogeneity in the group compromises the external validity of the study findings.

In terms of multivariate analysis methodology, I think simply a statement needs to be made about how you chose what univariate variables you’ve included into your multivariate model and which of the univariates were statistically significant in their own right. Given there is an imbalance between age, tumour classification, nodal stage and response group, you ought to perform multivariate analysis to adjust for the difference to be able to assert there is no difference in outcome between the two regimens or alternatively case matching on say the nodal status and tumour classification to demonstrate that after adjustment the two regimen shows more definitively the lack of difference in a neoadjuvant approach. I accept this may mean smaller study numbers and thus reduce the power of the study but this ought to be done to at least as supportive evidence or highlight uncertainty from the data. Otherwise right now whilst the overall results show no difference but then one ought to frame it such that the conclusion is limited by the imbalances in your groups.

No real concern about English language quality - will defer to editors for final proof reads.

Author Response

(The authors gave the same response as above.)

Round 2

Reviewer 2 Report

Thank you for taking the time to address all comments and accept your explanations and alterations.